

# On the origin of mitochondria: a multilayer network approach

Dérick Gabriel F. Borges[1], Daniel S. Carvalho[2], Gilberto C. Bomfim[3], Pablo Ivan P. Ramos[4], Jerzy Brzozowski[5], Aristóteles Góes-Neto[6,7], Roberto F. S. Andrade[1,8] and Charbel El-Hani[3,8]

[1] Institute of Physics, Federal University of Bahia, Salvador, Bahia, Brazil
[2] Institute of Biological Sciences, Federal University of Minas Gerais, Belo Horizonte, Minas Gerais, Brazil
[3] Institute of Biology, Federal University of Bahia, Salvador, Bahia, Brazil
[4] Gonçalo Moniz Institute, Oswaldo Cruz Foundation, Salvador, Bahia, Brazil
[5] Philosophy Department, Federal University of Santa Catarina, Florianópolis, Santa Catarina, Brazil
[6] Institute of Biological Sciences, Universidade Federal de Minas Gerais, Belo Horizonte, Minas Gerais, Brazil
[7] Graduate Program in Bioinformatics, Institute of Biological Sciences, Federal University of Minas Gerais, Belo Horizonte, Minas Gerais, Brazil
[8] National Institute of Science and Technology in Interdisciplinary and Transdisciplinary Studies in Ecology and Evolution (INCT IN-TREE), Salvador, Bahia, Brazil

Corresponding author
Aristóteles Góes-Neto,
arigoesneto@gmail.com

## ABSTRACT

**Background:** The endosymbiotic theory is widely accepted to explain the origin of mitochondria from a bacterial ancestor. While ample evidence supports the intimate connection of Alphaproteobacteria to the mitochondrial ancestor, pinpointing its closest relative within sampled Alphaproteobacteria is still an open evolutionary debate. Many different phylogenetic methods and approaches have been used to answer this challenging question, further compounded by the heterogeneity of sampled taxa, varying evolutionary rates of mitochondrial proteins, and the inherent biases in each method, all factors that can produce phylogenetic artifacts.
By harnessing the simplicity and interpretability of protein similarity networks, herein we re-evaluated the origin of mitochondria within an enhanced multilayer framework, which is an extension and improvement of a previously developed method.
**Methods:** We used a dataset of eight proteins found in mitochondria ($N = 6$ organisms) and bacteria ($N = 80$ organisms). The sequences were aligned and resulting identity matrices were combined to generate an eight-layer multiplex network. Each layer corresponded to a protein network, where nodes represented organisms and edges were placed following mutual sequence identity. The Multi-Newman-Girvan algorithm was applied to evaluate community structure, and bifurcation events linked to network partition allowed to trace patterns of divergence between studied taxa.
**Results:** In our network-based analysis, we first examined the topology of the 8-layer multiplex when mitochondrial sequences disconnected from the main alphaproteobacterial cluster. The resulting topology lent firm support toward an Alphaproteobacteria-sister placement for mitochondria, reinforcing the hypothesis that mitochondria diverged from the common ancestor of all Alphaproteobacteria.

Additionally, we observed that the divergence of Rickettsiales was an early event in the evolutionary history of alphaproteobacterial clades.

**Conclusion:** By leveraging complex networks methods to the challenging question of circumscribing mitochondrial origin, we suggest that the entire Alphaproteobacteria clade is the closest relative to mitochondria (Alphaproteobacterial-sister hypothesis), echoing recent findings based on different datasets and methodologies.

## INTRODUCTION

Mitochondria are the organelles mainly responsible for energy conversion processes and ATP synthesis in eukaryotes, amongst other functions. These processes represent important evolutionary acquisitions that have enabled an increase of complexity in eukaryotic cells (*Embley & Martin, 2006*; *Atteia et al., 2009*; *Roger, Muñoz-Gómez & Kamikawa, 2017*). The endosymbiotic theory is widely accepted to explain the origin of mitochondria and chloroplasts from an endosymbiotic ancestor (*Gray, Wolstenholme & Jeon, 1992*; *Gray, Burger & Lang, 1999*). Furthermore, the endosymbiotic origin of mitochondria is supported by the fact that this organelle has its own genome and apparatus for protein synthesis, shares similar features with prokaryotic cells, and has a division cycle independent from the cell (*Osteryoung & Nunnari, 2003*). During the evolution of eukaryotic cells, mitochondria went through multiple processes of DNA transfer to the eukaryotic cell nucleus (*Timmis et al., 2004*). Therefore, the mitochondrial genome is currently smaller than the symbiont genome from which it originated (*Adams & Palmer, 2003*).

While it is widely accepted that eukaryotic mitochondria have a close affiliation to Alphaproteobacteria (*Schwartz & Dayhoff, 1978*; *Andersson & Kurland, 1999*; *Gupta, 2005*; *Gray, 2012*), several studies attempting to establish the identity of this clade have presented divergent findings (*Schwartz & Dayhoff, 1978*; *Esser et al., 2004*; *Fitzpatrick, Creevey & McInerney, 2006*; *Atteia et al., 2009*; *Chang et al., 2010*; *Sassera et al., 2011*; *Abhishek et al., 2011*; *Rodríguez-Ezpeleta & Embley, 2012*; *Thiergart et al., 2012*; *Ferla et al., 2013*; *Carvalho et al., 2015*; *Wang & Wu, 2015*; *Esposti et al., 2016*; *Roger, Muñoz-Gómez & Kamikawa, 2017*; *Martijn et al., 2018*; *Fan et al., 2020*; *Muñoz-Gómez et al., 2022*). Thus, which extant bacterial group is most closely related to mitochondria still remains an open question.

To advance current knowledge on this regard, multiple methods for phylogenetic inference have been employed. However, contrasting results and opposing statistical support limit the accurate interpretation of mitochondrial origin. Clades proposed as sister groups in the literature include the entire Alphaproteobacteria (*Martijn et al., 2018*; *Muñoz-Gómez et al., 2022*) and its following subgroups: the Rhodospirillales order (*Schwartz & Dayhoff, 1978*; *Esser et al., 2004*; *Esposti et al., 2016*), Rhizobiales + Rhodobacterales (*Atteia et al., 2009*), the Rickettsiales order (*Fitzpatrick, Creevey & McInerney, 2006*; *Chang et al., 2010*; *Rodríguez-Ezpeleta & Embley, 2012*; *Fan et al., 2020*),

Rhodobacterales + Rhizobiales + Rhodospirillales (*Thiergart et al., 2012*),Anaplasmataceae + Rickettsiaceae (*Sassera et al., 2011*; *Wang & Wu, 2015*), as well as the entire Alphaproteobacteria clade except the Rickettsiales (*Carvalho et al., 2015*). As it is never possible to completely define and incorporate in a single statistical model all variables needed to describe an evolutionary process (a known bias phylogeneticists refer to as systematic error (*Kumar et al., 2011*)), and a broad consensual evidence for their utility in real-world datasets is still lacking (*Abadi et al., 2019*), alternative and complementary approaches to traditional phylogenetic inference can potentially contribute to what is known and help generate new hypotheses of mitochondrial evolutionary history.

In order to tackle this open issue, a diversity of methods for phylogenetic inference has been employed. Among the well-known features of current methods of large-scale phylogenomic inference is their dependence upon statistical models of sequence evolution, which aims at inferring an evolutionary process given a set of sequences (*Kumar et al., 2011*) based on sophisticated methods for evaluating model fit to the data, a step defined as model selection (*Luo et al., 2010*). Nevertheless, broad consensual evidence for its utility in real-world datasets is still lacking (*Spielman & Kosakovsky Pond, 2018*; *Abadi et al., 2019*; *Spielman, 2020*). As an alternative strategy, several authors (including most of us) have provided evidence that Network Science approaches based on Sequence Similarity Networks (SSNs) (*Bapteste et al., 2013*) help the discovery of relationships between taxa (*Atkinson et al., 2009*; *Andrade et al., 2011*; *Larremore, Clauset & Buckee, 2013*; *Corel et al., 2016*; *Chowdhary, Löffler & Smith, 2017*; *Solís-Lemus, Bastide & Ané, 2017*; *Góes-Neto et al., 2018*). Therefore, complex networks have been continuously adopted to explain even complex evolutionary processes, including but not restricted to, horizontal gene transfer, gene domain fusion, and gene or genome introgression (*Corel et al., 2016*; *Pathmanathan et al., 2018*; *Ocana-Pallares et al., 2019*).

Herein, we leveraged a previously introduced network method to construct a multiplex network from several mitochondrial proteins sequence data. Such a multi-dimensional set was employed to simultaneously investigate the structural topology of similarity networks inferred from these proteins, integrating signal from multiple orthologs, all supported by a bootstrap analysis to assess the confidence of the observed clustering patterns. Several contributions by some of us (*Góes-Neto et al., 2010*; *Andrade et al., 2011*; *Carvalho et al., 2015*; *Góes-Neto et al., 2018*, *Carvalho, Schnable & Almeida, 2018*) have shown that the original single-layer network version of this method is reliable and consistent. Indeed, it has been explicitly demonstrated that it produces results comparable to those obtained using distance, maximum parsimony, maximum likelihood, and Bayesian methods (*Góes-Neto et al., 2018*).

## METHODS

### Dataset and sequence alignment methods

We used the dataset produced by *Wang & Wu (2015)* for phylogenomic analysis of the origin of mitochondria, which includes sequences from 86 different organisms. We declined to include newly reported data in order to be able to carry out a direct comparison of their results with ours. This dataset consists of protein sequences encoded
by genes originally present in the mitochondrial genome but currently found in both mitochondrial and nuclear DNA. Among the proteins included in the dataset, we only considered the following orthologs: (i) *NADH dehydrogenase subunit 1* (Nad1); (ii) *NADH dehydrogenase subunit 4* (Nad4); (iii) *NADH dehydrogenase subunit 5* (Nad5); (iv) *NADH dehydrogenase subunit 6* (Nad6); (v) *NADH dehydrogenase subunit 9* (Nad9); (vi) *Cytochrome B* (Cob); (vii) *Cytochrome C oxidase subunit 2* (Cox2); (viii) *Cytochrome C oxidase subunit 3* (Cox3). These proteins were selected because they are encoded by genes that are evolutionarily well-conserved, with low nucleotide substitution rate (*Wang & Wu, 2015*), making them particularly useful for reconstructing ancient phylogenetic relationships. In other words, we tried to minimize the effect of very heterogeneous substitution rates referred to in the Introduction. In addition, these proteins are present in all six eukaryotic organisms included in the dataset: (i) *Phytophthora infestans* (Oomycetes, thereafter abbreviated as Phyto); (ii) *Marchantia polymorpha* (Marchantiopsida, a liverwort, abbreviated as March); (iii) *Mesostigma viride* (Charophyceae, a green alga, abbr. Virid); (iv) *Reclinomonas americana* (Jakobea, a free-living heterotrophic flagellate, abbr. Reclino); (v) *Hemiselmis andersenii* (Cryptophyceae, a unicellular alga, abbr. Hemi); (vi) *Rhodomonas salina* (another unicellular alga of the Cryptophyceae class, abbr. Salin). These protein sequences are also found in organisms of Alphaproteobacteria, Gammaproteobacteria, and Betaproteobacteria. Representatives from the latter two classes were used as external groups in our study.

Sequence alignments were carried out using Clustal Omega (version 2.1) (*Sievers et al., 2011*), available at https://www.ebi.ac.uk/Tools/msa/clustalo/, which was used to perform a global alignment applying default parameters. The protein identities obtained from the alignment files between organisms in the dataset were used to generate square identity matrices $W$. Each matrix element $w_{ij}$ represents the percentage of amino acid identity of two protein sequences $i$ and $j$, in other words, the fraction of amino acids in homologous positions relative to the total alignment length. If $w_{ij} \neq w_{ji}$, both elements are replaced by their arithmetic mean values to preserve the symmetry of each matrix $W$. Using these representations, we constructed weighted networks for each ortholog (see *Andrade et al., 2011*), in which nodes represent the organisms, while weighted edges correspond to the identity values of corresponding protein pairs.

## Network and multiplex construction

From a mathematical point of view, a multiplex network $M$ is defined as a multilayer network where all nodes are represented in each multiplex layer, even if it does not have a connection to any other node in that layer. $M$ is a pair of sets $(G,C)$, where $G$ is the set of layers $G = \{G_\alpha ; \alpha \subset \{1 \dots m\}\}$, where each layer is undirected and weighted, so that $G_\alpha (X_\alpha, E_\alpha)$ has the usual network shape. The set $C$ defined as:

$$C = \{E_{\alpha\beta} \subseteq X\alpha \times X_\beta; \alpha, \beta \in \{1 \dots m\}; \alpha \neq \beta, \} \tag{1}$$

is the set of interconnections between nodes of different layers $G_\alpha$ and $G_\beta$ with $\alpha \neq \beta$. The elements of $C$ are named "crossed layers" while the elements of each $E_\alpha$ are called

"intra-layer" connections of $M$ in contrast with the elements of each $E_{\alpha\beta}$ ($\alpha \neq \beta$) that are called "inter-layer connections" (*Boccaletti et al., 2014*).

Thus, the eight weighted networks described in the previous subsection were used to generate a single weighted multiplex with eight weighted layers. Within a layer obtained for a given ortholog, a node represents a specific organism that will also be represented in all other layers. The intra-layer weighted connections present in that layer represent the pair-wise protein identities among organisms. Besides that, within a multiplex each organism in any given layer is connected by an unweighted inter-layer connection to its counterpart in all the other seven layers. We remark that, when assembling a $L$-layer multiplex from its individual layers with $n$ nodes, it is necessary to define a single enumeration for its $nL$ nodes. We follow the usual convention, which starts by using the same number to identify a given node in all networks that will be a multiplex layer. Thus, a node that receives a number $i$ in the first multiplex layer will be present in all $L$ layers, being identified by $(\alpha-1)n + i$, $\alpha = 1, 2,\ldots,L$. For instance, if we assume a multiplex composed of three distinct layers of $n = 10$ nodes (proteins) each, then organism X represented by node 1 in layer 1 should be represented by node 11 in layer 2, by node 21 in layer 3, and so forth. The flow chart in Fig. 1 illustrates the main steps of the developed framework.

As briefly indicated above, when using this framework to analyze actual data, it may be happened that a node is not represented in a given layer, either because the organism it represents does not encode the corresponding protein, or because the protein sequence data are not available in the dataset. To overcome this limitation, when any of these conditions is observed for a given layer, the node representing this organism is included in the layer as an isolated node, which amounts to setting to zero its identity score with all the other sequences on that layer. Thus, each layer of the multiplex network showed an identical number of nodes, while the links of an isolated node are inter-layer connections.

A key step of the adopted procedure consists in obtaining, for each weighted network, a family of unweighted networks depending on a parameter $\sigma$, which give rise to a corresponding family of unweighted multiplexes. Indeed, useful phylogenetic information can be associated with the multiplex community structure evaluated at critical threshold values $\sigma_{th}$, following the same strategy originally introduced for the case of single layer network (*Andrade et al., 2011*). There it is argued that these values correspond to optimal choices between inter-community edge elimination (removal of noise) and intra-modules link preservation, the latter representing the actual signal that can be used to detect biologically informative communities. To achieve this in single-layer networks, it is necessary first to identify a small set of optimal values for $\sigma$. Such an identification can be made using the concept of network distance (*Andrade et al., 2008*), $\delta(\alpha, \beta)$, a measure of how dissimilar two networks $\alpha$ and $\beta$ are from each other. To better emphasize the meaning of this measure, from now on we will refer to $\delta(\alpha, \beta)$ as the dissimilarity between $\alpha$ and $\beta$. Proper threshold values of $\sigma$ can be found using the dissimilarity $\delta(\alpha, \beta)$, where $\alpha$ and $\beta$ represent the networks obtained at near threshold values, say $\sigma$ and $\sigma + \Delta\sigma$, which is denoted by $\delta(\sigma,\sigma + \Delta\sigma)$ (*Andrade et al., 2008*), (see Fig. 2). The network dissimilarity results exhibit sharp peaks of dissimilarity values the maxima of which we denote as $\sigma_{th}$, as indicated in Fig. 2. These peaks occur at, or immediately before, the occurrence of

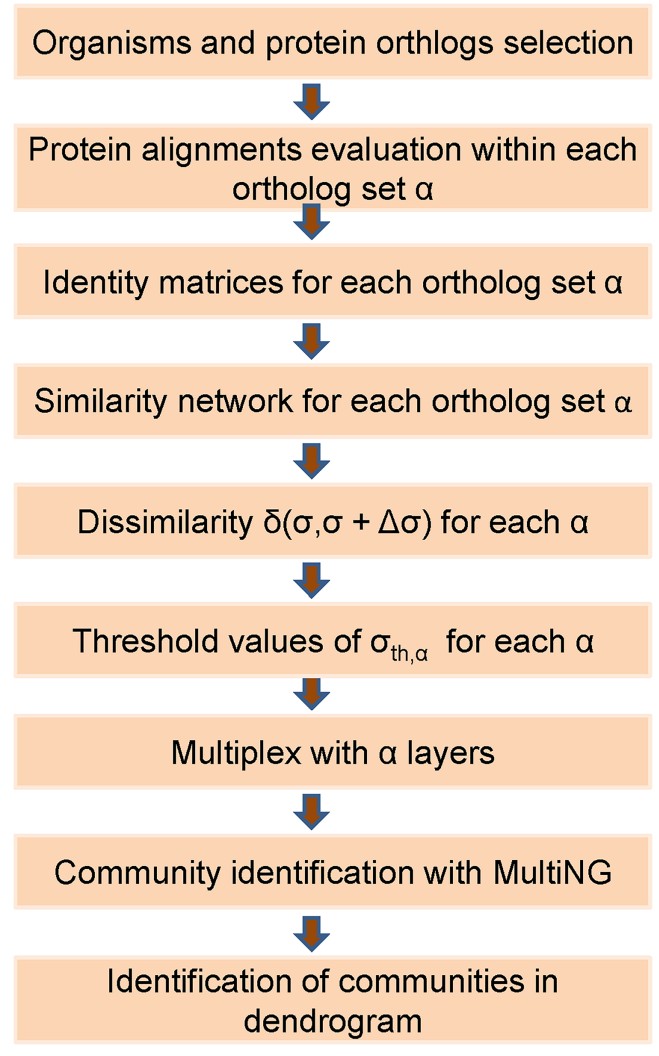

**Figure 1 Flowchart of the used frame based on the MultiNG algorithm to multiplex networks.** Sequence of steps required to obtain a dendrogram and community structure within the used approach. After the selection of organisms and protein orthologs, identity matrices are obtained from protein alignments (pair-wise or global) which allow for obtaining similarity networks. Dissimilarity measure $\delta(\sigma, \sigma + \Delta\sigma)$ is used to evaluate threshold values $\sigma_{th}$ for each ortholog, leading to the multiplex. Communities are identified using MultiNG and resulting dendrogram.

significant changes in the network's modular structure, which indicate its splitting into separate communities.

When working with a family of unweighted multiplexes, two strategies might be considered: (i) identify critical thresholds of $\sigma_{th,\alpha}$ in each layer $\alpha$; or (ii) evaluate a single $\sigma_{th}$ threshold for the entire multiplex. Given the complexity of the second approach, this work was limited to the first described procedure. In addition, for this type of study, the first approach leads to a better preservation of identity relationships between the pairs of sequences of the same protein, as discussed in *Borges & Andrade (2020)* for other types of networks.

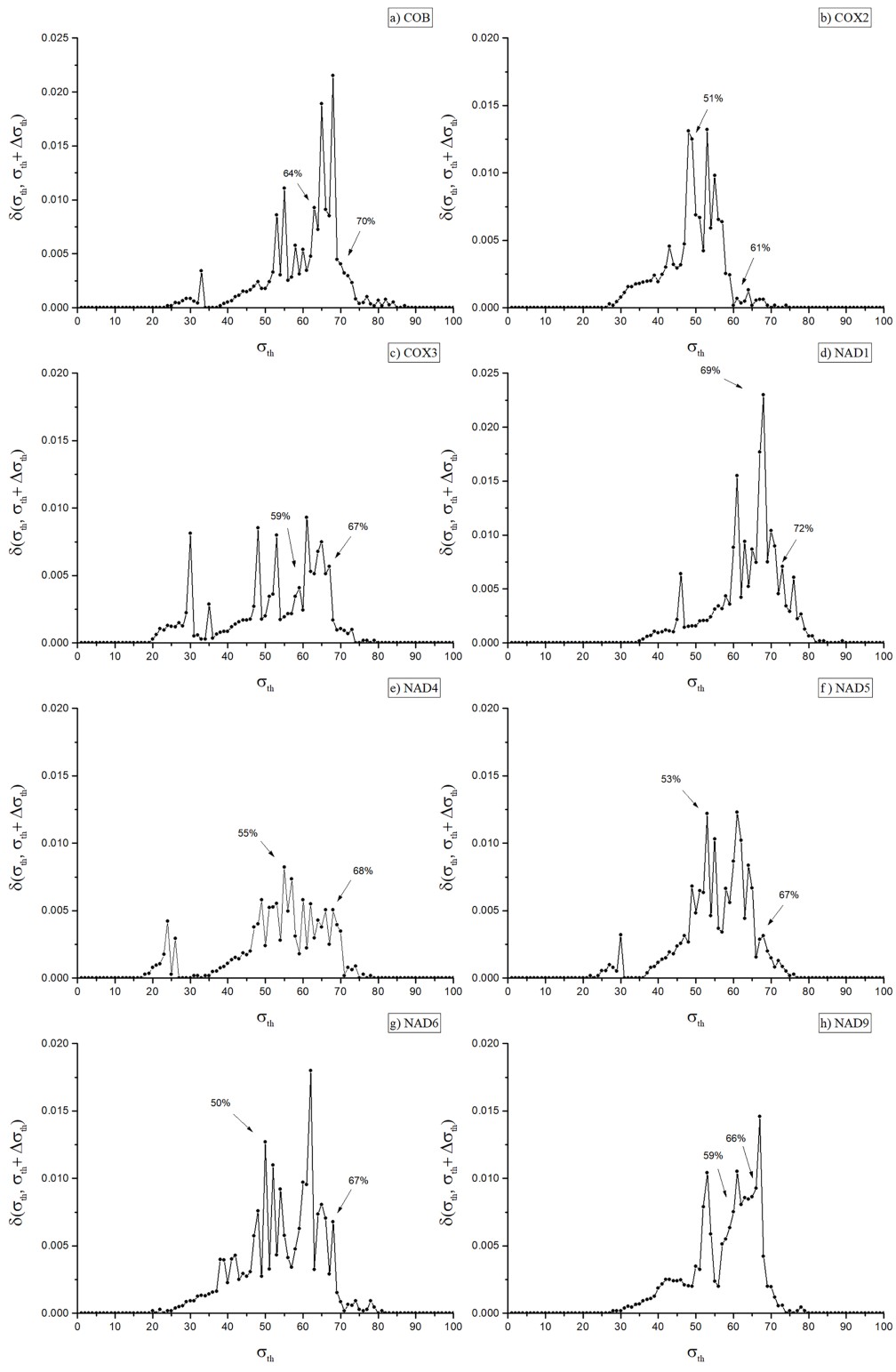

**Figure 2 Dissimilarity graph for the networks of the eight included orthologues.** Dissimilarity graph for the networks of the eight included orthologues. Arrows indicate the values of $\sigma_{th}$ used to obtain the dendrograms in Fig. 3 (Cob: 64%; Cox2: 51%; Cox3: 59%; Nad1: 69%; Nad4: 55%; Nad5: 53%; Nad6: 50%; Nad9: 59%) and in Fig. 1 of the Supplemental Material (Cob: 70%; Cox2: 61%; Cox3: 67%; Nad1: 72%; Nad4: 68%; Nad5: 67%; Nad6: 67%; Nad9: 66%).               

Once the set of $\sigma_{th}$ values are identified for each layer, adjacency matrices $A(\sigma_{th})$ are constructed using the following definition based on the values of $\sigma$:

$$A(\sigma_{th})_{ij} = \begin{cases} 1, & if \ w_{ij} \geq \sigma_{th}, \\ 0, & \text{otherwise} \end{cases}, \tag{2}$$

where $\sigma_{th}$ denotes a lower limit of the identity between protein sequences of two different organisms necessary to be inserted in network $A(\sigma_{th})$. As shown in *Borges & Andrade (2020)*, this procedure was successfully extended to the multiplex network. After generalizing Eq. (2), it is possible to obtain parameter families, selecting different $\sigma_{th}$ thresholds, to represent $\sigma_{th,\alpha}$ in each $\alpha$ layer.

## Community identification and dendrogram visualization

The Newman-Girvan (NG) algorithm (*Girvan & Newman, 2002*; *Newman & Girvan, 2004*) was one of the first proposed methods to identify the best modular partition of an unweighted single layer network described by its adjacency matrix of elements $A_{ij}$. The NG method consists of the consecutive removal of links by decreasing edge-betweenness centrality until all nodes are disconnected. The complete edge removal sequence can be represented as a dendrogram depicting the ramification process. The dendrogram starts with a single community $C$ on its left hand side and follows a series of bifurcation events. In each of them, a pre-existing community $Cin$ branches into smaller communities, until $n$ isolated leaves, which correspond to the number of network nodes, result on the right-hand side of the dendrogram. Besides that, at every step of the NG method, the modularity function $Q$ can be evaluated according to Eq. (3) as

$$Q = \frac{1}{2m} \sum_{ij}^{n} \left\{ A_{ij} - \frac{k_i k_j}{2m} \right\} \delta(C_i, C_j) \tag{3}$$

where $k_u$ is the degree of node $u$, $m$ is the number of edges in the network and $C_u$ denotes the community to which node $u$ belongs. The NG method was generalized to multi-layer networks in Multi-Newman-Girvan (MultiNG) (*Borges & Andrade, 2020*). It also computes the multiplex modularity function described by *Mucha et al. (2010)* within the GenLouvain framework. Here, we used MultiNG to identify communities in the multiplex networks. Dendrograms were constructed using Origin version 2015 (OriginLab, Northampton, MA, USA).

## Bootstrap resampling procedure

The bootstrap method used in this study was based on a previous implementation of a bootstrap procedure for the complex network approach (*Góes-Neto et al., 2018*). For each layer of the multiplex network, we generated 1,000 bootstrap replicates by resampling the identity matrix $W$ using the following bootstrap sampling scheme: First, one of the layers was chosen at random. Each value $wij$ of the layer's original identity matrix was divided by 100 and taken as the probability $p$ of success in a binomial distribution. Subsequently, $Y$ (the length of the sequence alignment for that layer) samples were drawn randomly with replacement from this binomial distribution, resulting in $k$ successes. Finally, the number

of successes was normalized and taken as the value of $\sigma_{tw}$ for that element, according to the formula $Wij = k \times 100/Y$. Except for the resampled layer, all the other layers are left undisturbed by this resampling procedure. The resulting resampled bootstrap multiplex network is then run through the same community-detection algorithm as above, using the original samples' threshold ($\sigma_{tw,\alpha}$) values to generate the corresponding adjacency matrices.

Of note, unlike traditional bootstrapping (*Felsenstein, 2004*), this method resamples over alignment scores rather than sequences themselves. However, since our method relies on resampling over a binomial distribution of similarity scores, one can reasonably expect it to emulate the behavior of similarity scores under the traditional method. It is expected that a greater similarity score between a pair of sequences would exhibit less variance under traditional resampling of the relevant sequences when compared to a lower similarity one. In fact, this is precisely the sort of behavior exhibited by the variance of similarity scores under the method used here (see Supplemental Information 4 in *Góes-Neto et al., 2018*).

Additionally, we have performed a traditional phylogenetic Bayesian inference of our matrix, concatenating all the eight protein datasets in the taxa ($n = 37$) in which all these eight proteins have been detected and publicly available. BI was performed using eight defined partitions (each one for each of the eight proteins) in MrBayes 3.2 (*Ronquist et al., 2012*) with two independent runs, each beginning from random trees with four simultaneous independent chains, for $1 \times 10^6$ generations, and sampling parameters every 1,000 generations. The first 25% of sampled trees were discarded as burn-in, while the remaining ones were used to reconstruct a 50% majority rule consensus tree and to calculate Bayesian posterior probabilities (BPP) of the clades.

Complete codes and all detailed information of each step of the MultiNG methods are publicly available for download from GitHub (https://github.com/randradeufba/MultiNG).

## RESULTS

### Sequence selection, construction of the mitochondrial protein network and identification of community structure

First, we selected a subset of proteins reported in the work of *Wang & Wu (2015)*. These eight proteins participate in key energetic processes and present broad evolutionary conservation. In the current dataset they were identified in 86 organisms: six eukaryotes, 72 Alphaproteobacteria, and in four members in each of classes Betaproteobacteria and Gammaproteobacteria. Within the Alphaproteobacteria, these proteins are distributed across nine orders, namely Rickettsiales ($n = 17$), Rhodobacterales ($n = 15$), Rhodospirillales ($n = 15$), Rhizobiales ($n = 13$), Sphingomonadales ($n = 4$), Caulobacterales ($n = 3$), Magnetococcales ($n = 2$), Kordiimonadales ($n = 1$), Parvularculales ($n = 1$), and one unclassified alphaproteobacterial representative (Table S1). Then, square identity matrices were generated following multiple alignment using Clustal Omega (Table S2), which were used as input to construct protein similarity Networks (PSN).

In the resulting similarity network for each protein, nodes correspond to organisms and there is a weighted, undirected edge between a node pair when their corresponding identity is larger than a predefined peak identity threshold $\sigma_{th}$. In practice, for each integer identity threshold in the range $\sigma_{th} \in [0, 100]$, a total of 101 networks can be generated and examined using many developed methods and measures (*Albert & Barabási, 2002*; *Newman, 2003*; *Boccaletti et al., 2006*; *Costa et al., 2007*). The choice of $\sigma_{th}$ thresholds is informed by inspecting the structure of the network at a given $\sigma$ and identifying the point immediately preceding the split of mitochondrial sequences from the remaining nodes in the network. Building upon the observation that PSNs can be used to inform studies of evolutionary relationships among taxa (*Atkinson et al., 2009*; *Corel et al., 2016*), and that peak $\sigma_{th}$ thresholds are driven by evolutionary branching processes (*Góes-Neto et al., 2018*), we subsequently investigated the structure of each individual protein network.

As shown in Fig. 2, in the eight disclosed PSNs dissimilarity values ranged from 50% (for Nad6) to 69% (Nad1), with mean dissimilarity of 58%. The remaining PSNs had peak identity values of 51% (Cox2), 53% (Nad5), 55% (Nad4), 59% (Cox3 and Nad9) and 64% (Cob). Once the optimal values for $\sigma_{th}$ were identified, an eight-layer unweighted multiplex network was constructed, each layer representing a protein network at its corresponding $\sigma_{th}$ threshold. Next, we used the inferred multiplex network to study in detail the modular structure and partitioning of the mitochondrial/bacterial protein sets and evaluate the evolutionary implications for mitochondrial origin.

## The eight-layer multiplex protein network supports the Alphaproteobacteria-sister placement of Mitochondria and the early branching of Rickettsiales

Initial analysis of the multiplex network aimed at evaluating its modular structure. For this, the MultiNG method (*Borges & Andrade, 2020*) was used, which expands upon the widely used Newman-Girvan method for community identification in single layer networks by successive elimination of links with largest betweenness centrality until all nodes are disconnected (*Newman & Girvan, 2004*). By estimating the modularity value for the multiplex network immediately before the branching out of mitochondria, here identified as Q = 0.383 using MultiNG (blue dotted line in Fig. 3 lower panel), one arrives at the best performing partition of the network. Converted into a dendrogram, the process of consecutive edge elimination during community identification likely reflects underlying bifurcation events that took place during the evolutionary history of the mitochondrial ancestor and its host (Fig. 3). Thus, since the community detection is performed, phylogenetic relationships among these different communities can be performed.

Our results showed a clear separation of communities grouping mitochondrial sequences from those harboring bacterial sequences, allowing to establish the relative positions of members of these groups (Fig. 3). Three main communities in the multiplex network could be resolved, named C1 to C3 in the dendrogram shown in Fig. 3. At the chosen values of $\sigma_{th}$, four organisms are completely isolated in all multiplex layers. For the purpose of clear identification, they appear close together in the dendrogram and named as

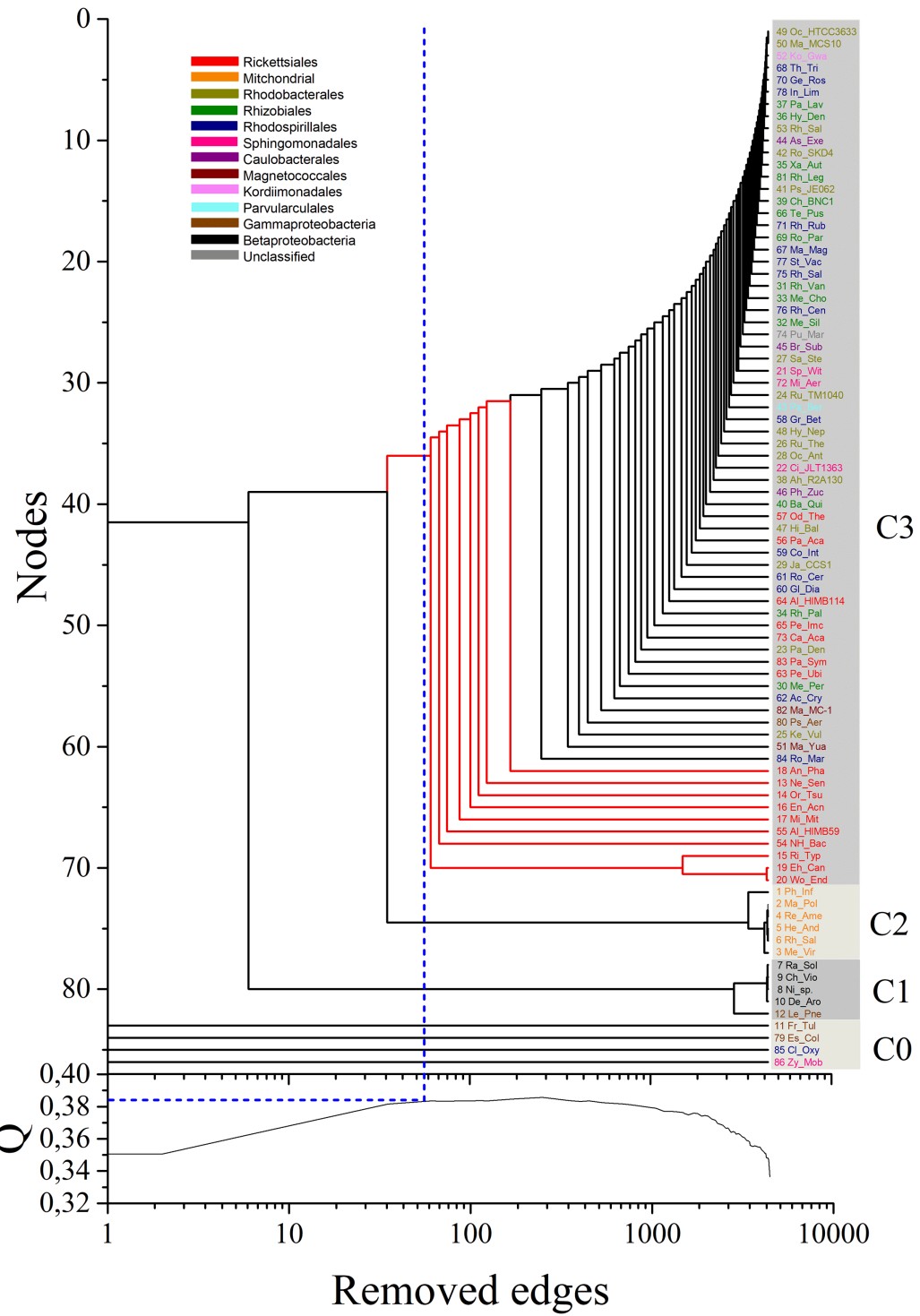

**Figure 3 Dendrogram obtained by the application of the MultiNG algorithm to the weighted multiplex network.** The dendrogram obtained by the application of the MultiNG algorithm to the weighted multiplex network depicts the three retrieved communities (C1, C2, and C3). Higher-order taxonomic information is indicated in the upper left legend. Q = 0.383 is the highest modularity value (blue-dotted line in lower panel) before the subsequent divergence within community C3 (blue-dotted line in the dendrogram).

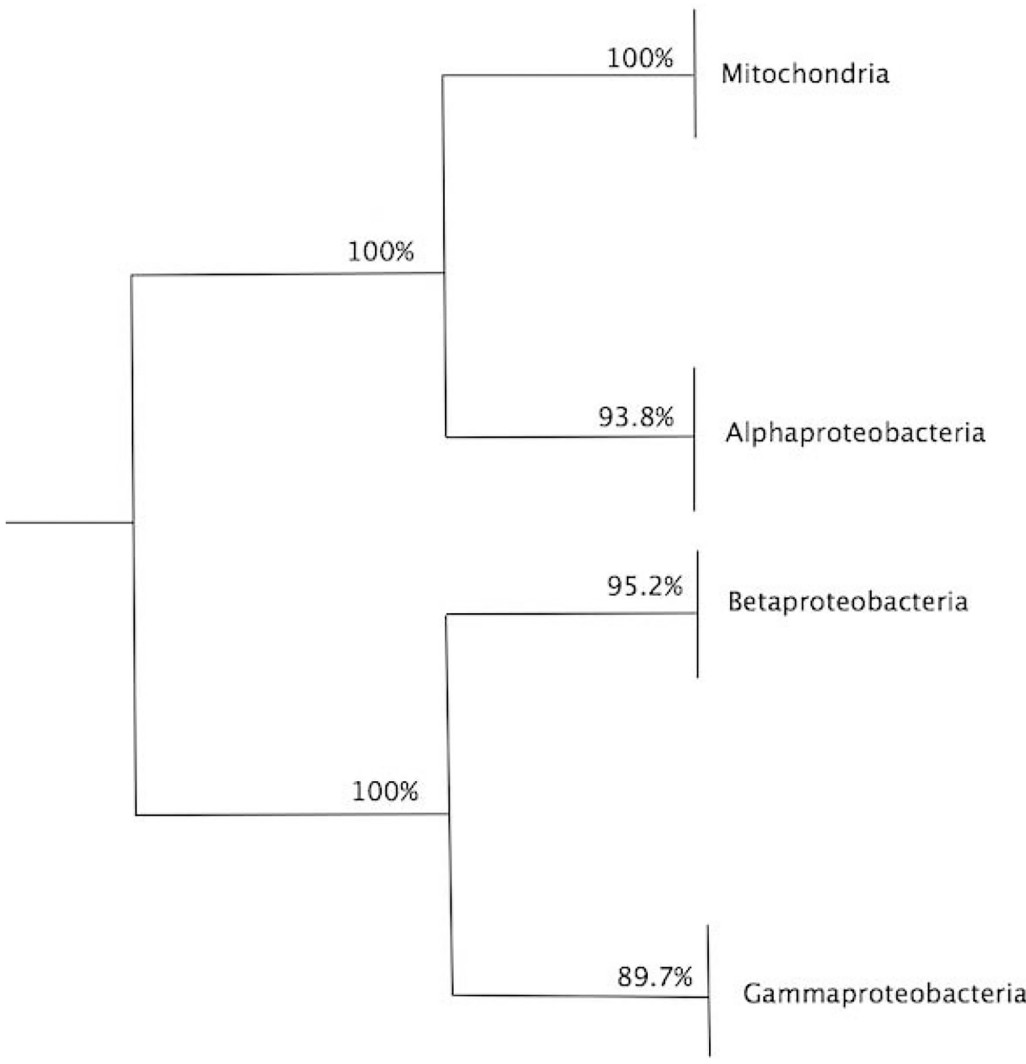

**Figure 4 Summary of bootstrap results for the dendrogram in Fig. 3.** Summary of bootstrap results for the dendrogram in Fig. 3, showing the values for clade, alphaproteobacteria, mitochondria and external group. A total of 1,000 bootstraps were used.

C0, although they actually do not form a cluster. Table S1 details the members of each identified community. Notably, these results support that the split of mitochondria and Alphaproteobacteria occurred prior to the separation and subsequent diversification of the inner alphaproteobacterial clades included during this analysis.

To assess the confidence of the obtained community partitioning, and extract further conclusions from each individual community, a bootstrap analysis with 1,000 rounds of random reshuffling of the similarity matrix was conducted, followed by manual inspection of members in each module. Community C2 grouped all eukaryotes with 100% bootstrap support; C3, the largest community, comprised 71 members and grouped all alphaproteobacterial sequences with 93.8% bootstrap support; and C1 contained members of Gammaproteobacteria and Betaproteobacteria, employed as outgroups (Fig. 4).

The Alphaproteobacterial-sister placement of mitochondria was supported in 100% of the bootstrap samples, further reinforcing the robustness of the results (Figs. 3 and 4).

By analyzing the bifurcation events that took place within community C3, we identified the early branching, as revealed by the sequential removal of edges in the multiplex, of 10 out of the 17 Rickettsiales sequences included in the dataset (red branches in Fig. 3). This provides strong support toward the divergence of Rickettsiales before the diversification of the remaining alphaproteobacterial subgroups. It is noteworthy to observe that almost all branching events within community C3 occur in the form of isolated organism leaving the group, without giving rise to any relevant sub-community. This may be related to the fact that, as our main focus lies on the early branching from mitochondria community C2, for all α layers we selected values of $\sigma_{th\ \alpha}$ that could emphasize this particular event. Because of that, C1 and C2 clearly branch out after a relatively small number of edge elimination within the MultiNG approach, and the further branching events within C3 become somewhat blurred. So, to further exemplify the ability of MultiNG, we conducted a second analysis where we considered increased values of $\sigma_{th\ \alpha}$ (see Fig. 2). As expected, it is not able to indicate the branching events leading to C1 and C2, but it does show sub-communities within C3. As these events do not constitute the focus of our current study, we refrain to discuss the biological consequences of these results. The resulting dendrogram is shown in Figure M1. Furthermore, just for comparison with an established tree-based phylogenetic method, we have also run the analysis performing a Bayesian Inference in Mr.Bayes using the best-fit model of protein according to AIC (LG+I+G+F with 100.0% of confidence interval), empirically calculated by ProtTest 3.2 (*Darriba et al., 2011*), and the results were similar to those proposed by *Wang & Wu (2015)* (supporting the Rickettsiales-sister hypothesis), and; thus, differently from what we have retrieved from our Multilayer network approach (Fig. S2). Moreover, our MultiNG approach is scalable and faster than traditional phylogenetic Bayesian inference (Fig. S1).

## DISCUSSION

Uncertainties surrounding the evolutionary origin of mitochondria remain in the literature, with different studies reaching conflicting conclusions. This may be driven by the diversity of employed methods, limitations thereof as well as heterogeneity of sampled taxa. Difficulties for inferring the positioning of mitochondria within the Alphaproteobacteria clade also stem from variation in the composition of mtDNA (with genes enriched in A+T and proteins enriched in amino acids specified by A+T-rich codons) and in the evolutionary rates of proteins of mitochondrial origin. All combined, these biases can create phylogenetic artifacts that lead to mitochondrial misgrouping with rapidly evolving alphaproteobacterial clades, a phenomenon known as long-branch attraction (*Bergsten, 2005*; *Roger, Muñoz-Gómez & Kamikawa, 2017*). Approaches to avoid these biases have already been proposed, such as the use of nuclear-encoded mitochondrial genes (*Fan et al., 2020*). Nonetheless, such mitigation strategies still produced conflicting hypotheses, despite a strong consensus that mitochondria evolved from an ancestor related to the order Rickettsiales (Rickettsiales-sister hypothesis) (*Fitzpatrick, Creevey & McInerney, 2006*; *Williams, Sobral & Dickerman, 2007*; *Chang et al., 2010*; *Sassera et al.,*

*2011*; *Rodríguez-Ezpeleta & Embley, 2012*; *Wang & Wu, 2015*; *Fan et al., 2020*). An alternative hypothesis contends mitochondrial descent from a free-living group within the Alphaproteobacteria (Caulobacteridae-sister hypothesis) (*Schwartz & Dayhoff, 1978*; *Esser et al., 2004*; *Atteia et al., 2009*; *Abhishek et al., 2011*; *Thiergart et al., 2012*; *Esposti et al., 2016*). In 2018, *Martijn et al. (2018)* suggested, for the first time, that mitochondria branched out from a proteobacterial clade before Alphaproteobacteria divergence, which would make the entire alphaproteobacterial class a sister group to mitochondria (Alphaproteobacteria-sister hypothesis). And, very recently, *Muñoz-Gómez et al. (2022)* had developed a new model of protein evolution (MAM60 + GFmix) and applied it to an expanded dataset of both mitochondrial proteins of alphaproteobacterial origin and new metagenome-assembled genomes (MAGs), strongly supporting the Alphaproteobacteria-sister hypothesis initially suggested by *Martijn et al. (2018)*.

In order to cast light on this ongoing evolutionary debate, we leveraged a previously published dataset composed of nuclear- and mitochondrial-encoded proteins (*Wang & Wu, 2015*) and, using a complex network approach, we disclosed a protein similarity network (PSN) modeled as an eight-layer multiplex in which community identification was performed. This allowed recapitulating the evolutionary history of these sequences. We previously showed that PSNs offer an alternative to traditional tree-based phylogenetic methods, producing results analogous to those obtained with widely used techniques in this field (based on distance, maximum parsimony, maximum likelihood, and Bayesian inference) (*Góes-Neto et al., 2018*). Furthermore, PSNs are *a priori* less constrained and can provide additional information beyond traditional phylogenetic methods, also allowing the elaboration and testing of a higher number of evolutionary hypotheses (*Bapteste et al., 2013*; *Harel et al., 2015*; *Watson et al., 2019*).

PSNs are constructed using as input a set of orthologous proteins, and phylogenetic inference can be achieved using simple networks. Alternatively, a set of multiple orthologous proteins can be simultaneously analyzed, yielding a multiplex network, the approach used in the present work. Taking advantage of the signal contained in the modular structure of these networks, communities can be recovered, and phylogenetic relationships can be inferred from the partition of these networks into communities (*Andrade et al., 2011*). In addition, multilayered structures adequately capture the complexities of real-world systems with greater accuracy compared to simple networks, as the limited complexity modeled in single-layer networks can lead to misleading interpretations (*Boccaletti et al., 2014*; *Kivelä et al., 2014*; *De Domenico et al., 2016*).

Our results support the hypothesis that mitochondria share a common ancestor with the entire Alphaproteobacteria clade (Alphaproteobacteria-sister hypothesis). Interestingly, this is at odds with the work of *Wang & Wu (2015)*, although our dataset was derived from their study, which employed maximum likelihood and Bayesian methods to reconstruct phylogenetic trees. In that analysis, mitochondria were placed within Rickettsiales, forming a sister clade to Rickettsiaceae/Anaplasmataceae, with Holosporales as a sister group to both (*Wang & Wu, 2015*). These differences can be partly explained by methodological variations and by the composition of the dataset, considering a subset of the sequences reported in that study was used here. Our findings are in agreement with a

more recent work that attempted to mitigate artifacts associated with long branch attraction, compositional bias, and biased taxon sampling simultaneously (*Martijn et al., 2018*), as well as in *Muñoz-Gómez et al. (2022)*, using with both a highly extended dataset and the development of a new model of protein evolution. The bifurcation events evaluated in our network analysis also substantiate the hypothesis that mitochondria are closely related to the common ancestor of all Alphaproteobacteria, as originally proposed by *Martijn et al. (2018)*, and strongly supported by *Muñoz-Gómez et al. (2022)*, using 108 mitochondrial proteins of alphaproteobacterial origin, and novel MAGs from microbial mats, microbialites, and sediments.

We also examined in detail the divergence of Rickettsiales members, particularly given the proposition of a Rickettsiales-sister hypothesis for the origin for mitochondria. We identified that Rickettsiales sequences remained grouped within a community subtended mostly by Alphaproteobacteria representatives. With the consecutive removal of edges during the community identification procedure, most of Rickettsiales taxa were progressively disconnected from the remaining Alphaproteobacteria, recovering the early divergence of the Rickettsiales according to the topology proposed in previous alphaproteobacterial phylogenies, in which Rickettsiales ancestors branch out before other orders in most configurations (*Sallström & Andersson, 2005*; *Gupta & Mok, 2007*; *Thrash et al., 2011*; *Viklund, Ettema & Andersson, 2011*; *Le, Pontarotti & Raoult, 2014*). There is also evidence that Holosporales are not related to Rickettsiales (*Georgiades et al., 2011*; *Ferla et al., 2013*; *Szokoli et al., 2016*; *Muñoz-Gómez et al., 2019*). Altogether, these results expose uncertainties in the proposal of Rickettsiales as a sister group of mitochondria. Instead, our data confirm that this order is deeply rooted in Alphaproteobacteria, in spite of being the first alphaproteobacterial order to diverge, further strengthening the inference that mitochondria share a common ancestor with all alphaproteobacterial clades.

## CONCLUSIONS

Attempts to establish the closest relatives of mitochondria have reached conflicting results, despite the increasing availability of sampled taxa driven by high-throughput sequencing. Herein, we advance this debate by drawing on a multiplex network approach derived from a method that has been shown to generate results as reliable as those of other, widely used phylogenetic methods (*Góes-Neto et al., 2018*).

In regard to the scalability of the MultiNG procedure, we should recall that the algorithmic complexity of original NG algorithm is $m2n$, where $m$ and $n$ indicate the number of links and nodes in a single layer network, respectively. We have found out that our implementation of the MultiNG is consistent with the same scaling behavior, provided that $m$ and $n$ indicate the total number of inter- and intra-layer links and the total number of nodes, *i.e.*, the number of nodes in each layer times the number of layers (Figure S1). It is well known that the algorithmic complexity of the single layer NG algorithm can be reduced by using more efficient algorithms to evaluate the betweenness centrality, as already shown by *Brandes (2001)*. Our MultiNG implementation does not incorporate such procedure, but it is expected that its algorithmic complexity will also be reduced if the Brandes or even more efficient algorithm is implemented.

Hence, supported by robust statistical bootstrap resampling, our eight-layer network leverages the phylogenetic signal distributed among orthologs of Cob, Cox2, Cox3, Nad1, Nad4, Nad5, Nad6, and Nad9 proteins, and the detailed study of communities detected in this multiplex network firmly placed the entire Alphaproteobacteria clade as the sister group of mitochondria, and, thus, supporting the Alphaproteobacteria-sister hypothesis.

### Funding

Charbel El-Hani thanks CNPq (grant number 465767/2014-1) and CAPES (grant number 23038.000776/2017-54) for their support of INCT IN-TREE, to CNPq for support in the form of productivity in research grant (grant number 303011/2017-3), and to CAPES and UFBA for Senior Visiting Researcher Grant included in the CAPES-PRINT Program, which funds his stay in the Centre for Social Studies, University of Coimbra, Portugal (grant number 88887.465540/2019-00). Daniel S Carvalho thanks CAPES for the support in form of a scholarship (grant numbers 88887.364931/2019-00 and 88887.511110/2020-00). Roberto F S Andrade received the support of the National Institute of Science and Technology for Complex Systems (INCT-SC Brazil), and of the Brazilian agency CNPq through Grants No. 422561/2018-5 and No. 304257/2019-2. The funders had no role in study design, data collection and analysis, decision to publish, or preparation of the manuscript.

### Grant Disclosures

The following grant information was disclosed by the authors:
CNPq: 465767/2014-1 and 303011/2017-3.
CAPES: 23038.000776/2017-54.
UFBA.
University of Coimbra, Portugal: 88887.465540/2019-00.
CAPES: 88887.364931/2019-00 and 88887.511110/2020-00.
National Institute of Science and Technology for Complex Systems (INCT-SC Brazil).
Brazilian agency CNPq: 422561/2018-5 and 304257/2019-2.

### Competing Interests

Aristóteles Góes-Neto is an Academic Editor for PeerJ.

### Author Contributions

- Dérick Gabriel F. Borges conceived and designed the experiments, performed the experiments, analyzed the data, prepared figures and/or tables, authored or reviewed drafts of the article, and approved the final draft.
- Daniel S. Carvalho conceived and designed the experiments, performed the experiments, analyzed the data, authored or reviewed drafts of the article, and approved the final draft.
- Gilberto C. Bomfim conceived and designed the experiments, analyzed the data, authored or reviewed drafts of the article, and approved the final draft.

- Pablo Ivan P. Ramos conceived and designed the experiments, performed the experiments, analyzed the data, prepared figures and/or tables, authored or reviewed drafts of the article, and approved the final draft.
- Jerzy Brzozowski conceived and designed the experiments, performed the experiments, analyzed the data, authored or reviewed drafts of the article, and approved the final draft.
- Aristóteles Góes-Neto conceived and designed the experiments, analyzed the data, authored or reviewed drafts of the article, and approved the final draft.
- Roberto F. S. Andrade conceived and designed the experiments, analyzed the data, authored or reviewed drafts of the article, and approved the final draft.
- Charbel El-Hani conceived and designed the experiments, analyzed the data, authored or reviewed drafts of the article, and approved the final draft.

## Data Availability

The raw data is available in the Supplemental File.

## Supplemental Information

Supplemental information for this article can be found online at http://dx.doi.org/10.7717/peerj.14571#supplemental-information.

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
