# Peer review of "On the origin of mitochondria: a multilayer network approach"

_PeerJ, doi:10.7717/peerj.14571_

## Round 0.1 · original submission · Major Revisions

Dear Dr. Borges and colleagues:

Thanks for submitting your manuscript to PeerJ. I have now received three independent reviews of your work, and as you will see, the reviewers raised some major concerns about the research. One reviewer raised serious concerns about the research and recommended rejection; however, the other two reviewers are a bit more enthusiastic yet still raised some areas that need substantial improvement. I agree with the concerns of all reviewers.

I am affording you the option of revising your manuscript according to all three reviews but understand that your resubmission may be sent to at least one new reviewer for a fresh assessment (unless the reviewer recommending rejection is willing to re-review). Please address all of these concerns in your rebuttal letter.

I would like to see a parallel analysis of the taxa and datasets you utilize that performs a traditional phylogeny estimation. This will serve as a benchmark for comparing the network analysis to the standard approach in the field. Per the reviewers, there appear to be missing citations. Please also expand the description of network approaches in the methods and figure legends; consider a figure that explains network approaches step-by-step (flow chart). Your work must be 100% reproducible; e.g., descriptions of all software must be included in the Materials and Methods section, and all software and/or code must be provided.

Please use color more effectively in your figures, particularly for distinguishing the different taxonomic groups. Taxon codes need to be translated to species names.

Consider something like the supplements I made in a paper from a while back: see PMID: 23475938. There are example layouts for making the analyses relevant to where the field currently stands regarding the origin of mitochondria within the Alphaproteobacteria. Do something that will make the researchers in the field interested in your work; this should be a contribution that adds to the debate and draws attention to another approach that can be meaningful to an old debate that continually gets rekindled with the discovery of new taxa and, particularly, new ways to analyze genomic datasets.

Good luck with your revision,

-joe

Reviewer 1 ·

Basic reporting

General comments:
Borges et al., set to address the phylogenetic placement of the mitochondrial lineage using an alternative phylogenetic approach to those traditionally used, a multi-layer network approach. The authors conclude that mitochondria are sister to the entire Alphaproteobacteria clade in agreement with the more traditional study of Martijn et al. (2018). The manuscript is relatively well written and provides enough background information in the Introduction. There are however a few inaccurate statements and improper citations (see below). The figures are okay, but I wished the figure legends were expanded to guide readers not familiar with network approaches and their visualization and analysis. (It might also be very useful if the authors added a figure that explains their network approach step-by-step.) Although the network methods are described, the software used is not mentioned in the Methods section. The software/code is also not included in the supplementary material as far as I could see. This affects the reproducibility of the study.

Experimental design

In principle, I do not believe that network approaches are more useful or accurate than supermatrix-based phylogenomic approaches, as the authors appear to imply. I tend to see network approaches as exploratory methods that can offer insights that have to be interpreted in the context of other results. Phylogenetic studies, in general, should embrace a series of different approaches to assess for the robustness of the methods and data under analysis. I am skeptical about the validity of these results, as they may easily stem from biases in the data (see below). The authors should investigate this possibility. One way is to analyze the same dataset with different methods (which themselves rely on different assumptions).

Validity of the findings

I am afraid that I cannot recommend the publication of this manuscript as it is. In my opinion, the study requires to be improved substantially. I recommend two major improvements: (1) use a larger dataset (datasets from more recent studies can be used), and (2) analyze the datasets with alternative methods (other network approaches or supermatrix approaches such as those used in the cited literature) to properly assess the performance and robustness of the method favored.

Additional comments

Specific comments:
Abstract: The origin of mitochondria was a singular event. I thus find using the plural ‘origins’ quite confusing.
L47-49: The issue of whether mitochondria enabled the complexification of the eukaryotic cell is highly debated. Recent research has casted doubt on this assumption (see Lynch and Marinov, 2015, 2017).
L52-53: This statement is wrong or confusing as mitochondrial division is regulated by the eukaryotic host cell.
L54: Please remove the word ‘direct’ from the sentence as this process follows a rather indirect path.
L57-59: The wording here has to be improved to be more specific. Strictly speaking what is universally accepted is a close affiliation to the Alphaproteobacteria taxon.
L71: Roger et al. (2017) is a review paper. They do not take or endorse that position.
L82-83: Given such a strong claim, I would expect to see a larger set of references backing it up.
L100-119: The methodological strategy involved very few marker proteins, and no mitochondrial genomes that belong to jakobids or green plants (these are gene-rich and relatively slow-evolving mitochondrial genomes). This is, in my opinion, a serious problem that this study has, partially inherited from the somewhat outdated study by now of Wang and Wu (2015).
L120-128: Protein similarity distances are very problematic. How are multiple amino acid replacements accounted for? How are other well documented evolutionary phenomena (across-site and across-branch compositional heterogeneity) accounted for? Replying on simple protein similarity distances may introduce considerable biases.
L88-90: This is indeed a central assumption that, no doubt, is extremely problematic. It is unquestionably false and becomes an even bigger problem when addressing a deep phylogenetic event such as the origin of mitochondria.
L227-237: The phylogenetic placement of the mitochondrial lineage is a very hard phylogenetic problem to tackle. Because of this, sophisticated methods and the larger amounts of carefully selected data have to be used. Unfortunately, eight proteins inevitably provide very little evidence (phylogenetic signal) to address such an ancient evolutionary event. Recent studies have used much larger datasets.
The ‘mother’ dataset is also somewhat outdated. At least two more recent studies, Martijn et al. (2018) and Fan et al. (2020) use much larger and comprehensive datasets. There is even a newer preprint that uses a much-expanded dataset, Munoz-Gomez et al. (https://www.researchsquare.com/article/rs-557223/v1).
L228: Mention which genes have been transferred to the nuclear genomes of eukaryotes. Note that the nad genes are still encoded in many mitochondrial genomes and that cox genes are almost invariably encoded in the mitochondrial genomes of all eukaryotes.
L299: This approach was not used by Fan et al. (2020). In fact, Fan et al. (2020) argue against using this approach.
L304-308: The recent preprint by Munoz-Gomez et al., (https://www.researchsquare.com/article/rs-557223/v1) adds additional support to the Alphaproteobacteria-sister hypothesis with new models and data. This manuscript should accordingly be cited here.
L309-318: Resolution of the approach employed in thus study appears to be very weak, as only very large 'communities' are identified and discussed. The internal structures of these communities appear very scrambled and by no means reflect previously established relationships (Figs. 2 and 4). This observation further speaks of the inadequacy of the method used.
A comparison to other methods should at least also be performed in this study.

Reviewer 2 ·

Basic reporting

On the origins of mitochondria: a multilayer network approach written by Dérick Gabriel F. Borges et al. described entire Alphaproteobacteria clade is the closest relative to mitochondria using 8 reported protein sequences which encoded by genes that are evolutionarily well-conserved with low nucleotide substitution rate from 86 different organisms. The Authors applied the Multi-Newman-Girvan algorithm to evaluate community structure, and bifurcation events. The result is interesting, although the results reached conflicting conclusions from previous studies using similar data sets, in which mitochondria most likely originated from a Rickettsiales endosymbiont already residing in the host, but not from the distantly related free-living Pelagibacter and Rhodospirillales were suggested.

Experimental design

ew questions come up why this discrepancy is occurred.
In the report of 2015 Scientific Reports paper written by Wang and Wu, Pelagibacter is distantly related, while in this study Pelagibacter is included in Rickettsiales, although this manuscript provides strong support toward the divergence of Rickettsiales. If ignoring Pelagibacter sequence data, the similar conclusion of this divergence?
Wang and Wu strictly used 18 phylogenetically novel species, while the Authors used 80 species. Inclusion of close relative species may rather increase vias. If the Authors use the same set of sequences from18 phylogenetically novel species, still result is different?
29 slowly evolving mitochondria-derived nuclear genes were used the previous 2015 study, while the Author used only 8 reported protein sequences. This also may affect? The Author should provide distance graph for the networks of the rest of 21 genes at least in supplemental information.

Validity of the findings

Written in the above Experimental design

Additional comments

Machine learning methods have been applied to a broad range of areas within genetics and genomics. is there any possibility to apply the newly available machine learning methods?

Mitochondria is in cytosol. This living environment could be largely different from those in Alphaproteobacteria. Curiously thinking the very early stage before endosymbiotic gene transfer. In this case some protein sequences may be altered to fit the cytosol environment. Strict number of conserved gene sequence may fall in this pit hole.

Reviewer 3 ·

Basic reporting

Borges et al. introduce in this manuscript a multilayer network approach to recover evolutionary signal from distance matrices, building on their previous work (e.g. Andrade et al. 2011, Goes-Neto et al. 2018). They exemplify their methodology by re-analyzing data that were collected in order to improve our understanding of the origin of mitochondria. The manuscript is well written and the experimental design / methodology is well described. I have found their approach truly interesting but, or because of this, I have some comments and questions (see next).


1. I believe the entire manuscript would benefit from small changes emphasizing the developed methodological approach. I begin with comments on the Abstract and the Introduction. Then see the end of my comments in this section.

1.1 As I said, I find the methodology very interesting and I think it is worthy to be explored and developed. However, I believe that the mitochondria origin issue is a huge one, requiring the battery of complex and robust phylogenetic methods and, especially, the incorporation of novel/independent samples/lineages/data. In this sense, I don’t see a clear or novel contribution to our understanding of the origin of mitochondria. Therefore, I suggest two modifications in the Abstract:
A) Make it more explicit that the authors are presenting an extension and improvement of a previously developed methodology.
B) Soften the expression “we provide evidence” (line 41).


1.2 Also in Abstract, lines 35-37: the sentence is a bit odd. Please check.

1.3 Lines 77-78: I don’t believe that methods that are not intended to perform evolutionary reconstruction can potentially convey an accurate picture of mitochondrial evolutionary history. They may contribute to what is known or they may help generate new hypotheses but I don’t get how network analyses (as this is the case) could convey an accurate evolutionary history.

1.4 In fact, there is some repetition (back and forward) in lines 73-78 and 79-84. I suggest merging/reordering the content (taking into account the previous comment #3).

1.5 Lines 88-91: to move from alignments to similarity/identity matrices you can simply count identical amino acids or you can apply an evolutionary substitution model. Not necessarily, all amino acid substitutions are treated equally. It isn’t clear if you are talking in general (in which case is wrong) or in particular to your methodology. Please check.

1.6 Lines 92-97: the multilayer approach could be emphasized (e.g. it allows to integrate signal from multiple orthologs).


2. Comment on figures.

2.1 Figure 1. Please, increase font size in axes and, in particular, all labels.

2.2 Figure 2 (and Figure 4). If possible, increase the font size of dendrogram tip labels.

2.3 Figure 3. Change alphaproteobacteria to Alphaproteobacteria.

2.4. Figure 4 is a “zoom in” of Figure 2? This should be made explicit.


3. Comments on topics not addressed in the manuscript. They might be included (as short comments) in the text. For instance, they could improve the Discussion.

3.1 There are no comments on how easy or difficult it is to implement these methodologies. Is there a tool available? What is the computational burden/time to run these analyses? How easy would it be to use as an alternative or complement to “traditional phylogenetic methods”?

3.2 The authors in this and other manuscripts have used protein alignments for the construction of similarity matrices and then networks. Do they envisage the use of these methods on other types of data? This would be interesting, especially because other biological data that can be converted into similarity matrices are not analyzable by at least part of the traditional phylogenetic methods (I am thinking right now of similarity matrices that could be inferred from structural alignments and not based on linear sequences, for example).

Experimental design

1. Wang and Wu (2015) say “using a set of 29 slowly evolving mitochondria-derived nuclear genes that are less biased than mitochondria-encoded genes as the alternative ‘‘well behaved’’ markers for phylogenetic analysis”. It is not clear why you have sub-selected only eight genes (lines 108-112). Please check and comment.

2. Lines 123-126. Perhaps something is wrong. As I can understand, after the global alignment, you calculate the percentage of identical amino acids in pairwise sequence comparisons -and I assume constant length given the global alignment. In which condition, given this strategy, Wij could be different from Wji? Please clarify. I saw that in the past you have worked with BLAST-derived similarity values. Is this the case?

3. Apart from replication issues, it is not indicated if the code for the multilayer network analyses is available. In any case, is this a specific code? Is this implemented in software available to the academic community?

Validity of the findings

I have concerns regarding the results on the origin of mitochondria.

1. I have seen before new/fast approximations to bootstrap calculations (on posterior matrices instead of original alignments) so I don’t have an issue with this. However, as we know, good bootstrap support does not mean that the node is correct (for example, as discussed by the authors, we can find high support to a node due to LBA artifact).

2. As the authors discuss, Rickettsiales appear as “branching basal” in the Alphaproteobacteria C2 cluster. However, the C2 cluster is far from being “resolved” and, for instance, Rickettsiales are not part of a different subcluster. First question: could this be improved if another sigma is determined by looking for changes/splits in the network topology inside Alphaproteobacteria? It is hard to feel comfortable with discussion lines 340-352 when there is no further clustering in C2. In this sense, Figure 4, as far as I can understand, is only a “zoom in” of Figure 2. Could it be improved in the sense of reporting additional results regarding Alphaproteobacteria?

3. Rickettsiales appear as early diverging but they are not a defined cluster. Second question: does the method have enough power to be able to locate mitochondrial sequences inside Alphaproteobacteria when it is an entire cluster? As I mentioned, the 100% bootstrap value could mean only support to the close relationship of mitochondria and Alphaproteobacteria compared to the outgroup. Third question: could you think about a way to test the second question?

4. In general, I would feel more confident/open about the mitochondria+Alphaproteobacteria relationship (instead of a mitochondira+Rickettsiales clade as many references support) if Rickettsiales were located in their own community o as early diverging but with issues #2 and #3 addressed. As I mentioned, while lines 328-352 are well written, I am not very confident about the validity of the specific results of the mitochondria origins.

Additional comments

1. line 22: “The” is in bold.

2. line 110: there is an extra parenthesis.

3. line 136: check “while and the elements”.

4. line 153: check “to because or protein sequence”.

5. lines 155-156: please check “which amounts to setting to zero its with identity score with all the other sequences on that layer.”

6. line 167: please check “(Fig. 1).*”.

7. line 179: change Eq. (2) to (eq.3) (standardize the format for equation numbering).

8. line 197: check “the multiplex the modularity”.

9. lines 265-266: check “bifurcation events… history of the mitochondrial ancestor and its host”. I would talk about mitochondria and its ancestor. Using “host” does not refer to the common ancestor of mitochondria and bacteria.

---

## Round 0.2 · Major Revisions

Dear Dr. Borges and colleagues:

Thanks for revising your manuscript. One reviewer was generally satisfied with your revision (as am I). Great! However, there are some more issues to address. Please address these ASAP so we may continue with your manuscript.

Best,

-joe

Reviewer 3 ·

Basic reporting

Borges et al. introduce in this manuscript a multilayer network approach to recover the evolutionary signal from distance matrices, building on their previous work (e.g. Andrade et al. 2011, Goes-Neto et al. 2018). They exemplify their methodology by re-analyzing data that were collected in order to improve our understanding of the origin of mitochondria. Considering the initial submission, the manuscript has improved and many of my comments have been incorporated. In this regard, I found the incorporation of the additional sigma value and the results and discussion of the community within Alphaproteobacteria especially enlightening. I thank the authors for this.

However, I maintain my concerns on some points that I consider critical. See "Experimental Design".

Regarding the figures, the legends should be improved even more. In particular, Figure 3 does not indicate the meaning of the blue or red lines, nor which table to go to look for the identity of the taxa labels. The legend should be more self-contained and less dependent on the main text. In addition, I suggest indicating with some kind of code (perhaps adding color elements), the higher order taxonomic information ("Taxonomic information/mitochondrial origin" column in Supplementary Table 1). It is the main figure with the results of the manuscript, so it would improve the manuscript’s impact if the figure is more informative (and eye-catching).

Experimental design

1. The authors have indicated in the rebuttal letter that they have used SCANNET (Barros-Junio et al 2018) to carry out the analyses. However, I have not found in the text of the new version this tool mentioned (and this paper is not included in the references, either), nor other finer details about the options used to be able to reproduce this analysis and results.

2. This type of analysis is mainly exploratory (I do not consider it to replace phylogenetic reconstruction methods). In particular, it could be useful for the use of other types of data not analyzable by traditional phylogenetic methods, and this is not explored in the new version of the manuscript either (although in fact the authors indicate that my earlier suggestion to do so was incorporated). The authors also do not incorporate information on how simple/complex, computationally expensive or not, it is to perform these analyses (compared to phylogenetic inference methods). As they only use 8 genes (even if they justify their choice) it is difficult to understand whether it is feasible to use this tool with other types of data, or with data of the same type but larger (because in general, we do not work with only 8 orthologues). I would really like to see some discussion to understand the feasibility of being able to use this approach in other contexts (or how many evolutionary hypotheses can be addressed -line 567).

3. The authors incorporated a Bayesian phylogenetic analysis on their 8-orthologs dataset, obtaining, however, a similar result to that of Wang and Wu (2015). Comments:
3.1 The methods should go in the corresponding section, and should better detail what has been done (e.g. the characteristics of the Metropolis-coupled Monte Carlo Markov Chains, whether the convergence of the estimated parameters was reached, and the burn-in percentage).
3.2 In the results section, a few more words should be said about what the result of the comparison of both methods is (and see next point).
3.3 Although the authors used 86 taxa, only 37 terminal branches and labels are visible in the “MrBayes tree”. It is not understood then why it is not exactly the same data set.
3.4 The supplementary figure must be renamed (there are two supplementary figures number 1)

Validity of the findings

See my comment above ("Experimental design")

Additional comments

Besides the general format problem in the pdf of version 1, please have a look at additional small details along the text. For example,
1. An asterisk in line 280
2. Replace "Fig. 2" with Figure 2 in line 281
3. Some end points are missing (e.g. lines 530 and 548)

---

## Round 0.3 · accepted · Accept

Dear Dr. Borges and colleagues:

Thanks for revising your manuscript based on the concerns raised by the reviewer. I now believe that your manuscript is suitable for publication. Congratulations! I look forward to seeing this work in print, and I anticipate it being an important resource for groups studying mitochondrial origins. Thanks again for choosing PeerJ to publish such important work.

Best,

-joe

Reviewer 3 ·

Basic reporting

Borges et al. introduce in this manuscript a multilayer network approach to recover the evolutionary signal from distance matrices, building on their previous work (e.g. Andrade et al. 2011, Goes-Neto et al. 2018). They exemplify their methodology by re-analyzing data that were collected in order to improve our understanding of the origin of mitochondria. They suggest that the entire Alphaproteobacteria clade
is the closest relative to mitochondria (Alphaproteobacterial-sister hypothesis), echoing recent findings based on different datasets and methodologies. Considering the initial submission, the manuscript has improved significantly. The authors have been really successful in clarifying the content and in making the manuscript more reader-friendly. Congratulations on this.

Experimental design

The experimental design is clear. The methodology and scripts have been all provided.

Validity of the findings

The findings are supported and well discussed.